# The Knowledge and Use of the International Classification of Functioning, Disability and Health (ICF) Framework in Athletic Training

**DOI:** 10.3390/ijerph20075401

**Published:** 2023-04-04

**Authors:** Nathaniel J. Millet, Alison R. Snyder Valier, Lindsey E. Eberman, Matthew J. Rivera, Zachary K. Winkelmann

**Affiliations:** 1Department of Exercise Science, University of South Carolina, Columbia, SC 29208, USA; 2School of Osteopathic Medicine, Department of Athletic Training, A.T. Still University, Mesa, AZ 85206, USA; 3Department of Applied Medicine and Rehabilitation, Indiana State University, Terre Haute, IN 47809, USA

**Keywords:** patient-centered care, sports medicine, professional development

## Abstract

In 2015, the Strategic Alliance adopted the International Classification of Functioning, Disability, and Health (ICF) as the disablement model framework for delivery of and communication about patient care in athletic training. The purpose of this study was to examine athletic trainers’ familiarity, knowledge, application, and implementation of the ICF framework. We used a cross-sectional online survey with 185 athletic trainers (age = 35 ± 9 y), which included 32 items focused on familiarity, knowledge, application, and implementation of the ICF framework. Most participants (*n* = 96, 51.9%) reported never learning about the ICF framework. During the knowledge assessment, participants scored 4.3 ± 2.7 out of 8, which is equivalent to 53.7%. For the sorting assessment, participants scored 10.9 ± 3.9 out of 18, which is equivalent to 60.5%. On the implementation matrix, the most frequently reported ICF tasks elicited by the athletic trainers included neuromusculoskeletal and movement, structure related to movement, and mobility. The most common ‘never elicited’ ICF tasks included voice and speech, sexual orientation, and structures related to genitourinary and reproductive system. Deficits related to the ICF framework exist. Athletic trainers reported low implementation across all ICF categories. The decision to not elicit information on these areas of health may reduce the ability to provide patient-centered healthcare.

## 1. Introduction

According to the Commission on Accreditation of Athletic Training Education (CAATE) 2020 Standards, the core competency of patient-centered care has five key standards for professional athletic trainers, which are as follows: (1) advocate for the health needs of clients, patients, communities, and populations; (2) analyze the impact of health literacy and social determinants of health on patient care and outcomes to determine healthcare strategies that empower patients and improve outcomes; (3) incorporate patient education and self-care programs to engage patients and their families and friends to participate in their care and recover; (4) communicate effectively and appropriately with clients/patients, family members, coaches, administrators, other healthcare professionals, consumers, payors, policy makers, and others; and (5) use the International Classification of Functioning, Disability, and Health (ICF) as a framework for delivery of and communication about patient care [1]. The use of disablement models such as the ICF framework allow clinicians to focus their treatment on the unique needs of each patient [2]. Disablement is experienced differently among individuals; however, the theme of each person’s experience can fit into four disablement components: impairments, functional limitation, disability limitation, and quality-of-life changes, thus making it a key perspective for patient-centered care delivery [3]. In a published application of the ICF model to athletic training, an adolescent patient participating in basketball sustains a distal radius fracture (health condition), which impacts her range of motion and mobility (body functions and structures impairments), limiting her ability to hold a pencil in class and dribble the basketball (activity limitations) and thus leading to problems with homework and participating in basketball practice (participation restrictions) [4]. In addition, a holistic lens of the ICF model explores environmental factors such insurance status and personal factors such as age, gender, and sexual orientation [4]. 

There are several forms of disablement models, including the World Health Organization (WHO) Classification, the Nagi Model, the National Center for Medical Rehabilitation Research (NCMRR) Model, and the ICF framework. In 2008, the National Athletic Trainers’ Association (NATA) determined the need to select a disablement model for athletic trainers to implement in the profession [5]. In 2015, the Strategic Alliance, including the NATA and the CAATE, adopted the ICF framework [6]. The ICF framework has been used as a tool in healthcare and has allowed clinicians to recognize biases, improve patient-centered outcomes, and facilitate helpful conversations with patients [7]. Athletic trainers can use the ICF framework to provide a means for delivering detailed injury evaluation, examine the effects of an injury on patients’ health-related quality of life, and treat their patients with a holistic approach [8].

Previous research from 2010 exploring the disablement process in physically active individuals with musculoskeletal injuries identified common limitations within the patient such as pain, decreased motion and muscle function, instability, skill performance, daily actions, maintaining a position, and quality of life through the lens of fear, energy, mood, and relationships [3]. These limitations describe the full perspective, from limitations at the body systems level to the impact on social roles, of how the injury is experienced for the patient. The use of patient-reported outcome measures has been used as a mechanism to explore the disablement model when assessing topics such as health-related quality of life, although multiple patient-reported outcome measures need to be utilized to capture multiple disablement domains and health dimensions [2]. Unfortunately, the use of patient-reported outcome measures continues to be infrequent [9,10,11]. The athletic training profession has been slow to integrate the ICF framework into clinical practice. In a recent study exploring key components and viewpoints of patient-centered care, a large majority of athletic trainers reported that they did not consider the ICF framework a necessary component of patient-centered care [12]. When following up with these participants, when they were asked why they did not consider the ICF part of patient-centered care, many individuals reported simply not knowing what the ICF framework was or how to use it [12]. Previous researchers remarked that if clinicians and researchers understand the concept of disablement, then treatment outcomes can be measured in areas patients identified as important [3]. We can presume that if athletic trainers do not fully comprehend the ICF framework and its use to support patient care, this will lead to a lack of implementation in their clinical practice. Therefore, the purpose of this study was to examine athletic trainers’ knowledge and implementation of the ICF framework. 

## 2. Materials and Methods

To examine athletic trainers’ knowledge and implementation of the ICF framework, we designed a cross-sectional study using a web-based survey. The Indiana State University Institutional Review Board deemed this study exempt.

### 2.1. Participants

The study was sent to 7000 potential participants who were members of NATA and opted into the survey research database. 

### 2.2. Instruments

To explore our research question, the research team developed a multi-item survey containing six demographic questions and three instruments focused on their knowledge, application, and use of the ICF framework (Appendix A). The first instrument (15 items) explored the participants’ background in relation to the ICF framework, including their familiarity, previous learning experiences (select all that applied), and 8 knowledge assessment questions that asked each participant to select the best answer (or ‘I do not know’ if they were unsure, rather than guessing) relative to where the example fit into the ICF framework components. In the second instrument, the participants completed a sorting matrix to apply a list of pre-selected category statements (*n* = 18) from the World Health Organization into the 6 major categories of the ICF framework. In the third and final instrument, the participants shared, in terms of their patient load, and when relevant to the patient’s condition, how often they elicited and/or documented information on the functioning and disability of a patient specific to the 5 major categories of the ICF framework. For each major category, a short list was created by extracting the tasks from the ICF Checklist 2.1 from the World Health Organization [13]. The participant shared if they elicited/documented each task on a 5-point Likert scale (never; some patients; about half of my patients; most of my patients; always). If the participant selected never for any of the tasks in the major category, a follow-up item with 5 possible responses was presented, asking them why they did not elicit/document that information. 

To validate the survey, a panel of experts (*n* = 3; 2 women and 1 man) who have researched and presented on the ICF framework were called upon. The expert started by marking each question as “needs attention” or “sufficient as written”, with detailed feedback on each question in the survey which they thought may need revisions. After that, the primary and senior investigator completed the revisions based on their feedback. Next, the research team sent a copy of the revised survey along with a content analysis rubric. The content analysis rubric asked each expert to score each item in the survey based on relevancy and clarity on a 4-point Likert scale (1 = not relevant, 4 = very relevant; 1 = not clear, 4 = very clear). The scores from the panel of experts were inputted into a data sheet to analyze the relevancy score and clarity score ranging from 0 to 1, with 1 indicating consensus amongst the panel. After the experts’ analysis, the survey received a content validation index (CVI) score of 0.95 for clarity. On average, an acceptable CVI score for relevancy is greater than or equal to 0.9 [14], which we exceeded. Overall, the survey achieved content validity (CVI; relevance = 1.0, clarity = 0.95).

### 2.3. Procedures

The research team collected the data electronically using an anonymous web-based survey (Qualtrics, Provo, UT). Recruitment emails were sent once a week beginning in May 2022. The initial emails were sent on 17 May 2022, then reminders were sent 24 May 31 May and 7 June. The same process was repeated for four weeks, beginning in September 2022. The initial emails were sent 6 September, then reminder emails were sent 13 September, 20 September, and 27 September. The same sample (*n* = 7000) was used for both data collection periods to capture those who never started or had unfinished responses. After clicking on the link, an inclusion criteria question was presented to ensure the respondent was a certified athletic trainer, followed by electronic informed consent. The participants then proceeded with the survey, with the option to stop and exit the survey at any time. 

### 2.4. Data Analysis

Data were collected and transferred from the web-based survey platform into a custom spreadsheet for data cleaning. We analyzed the data using SPSS 28 (IBM Corporation, Armonk, NY, USA). We performed descriptive statistics (mean, standard deviation, and frequencies) for all items. To measure the knowledge and background of athletic trainers in the ICF framework, we created a knowledge assessment score. The total number of correct answers out of all the questions asked was translated into a percentage score per participant for both the sort matrix and the definition knowledge-based questions. The knowledge assessment was out of 8 questions and the sort matrix was out of 18. When considering years of experience, we classified and compared the groups with 0 to 7 years of experience, given the adoption of the ICF in 2015, to those with more than 7 years of experience using a Mann–Whitney U non-parametric statistic for the knowledge assessment score. Significance was set at *p* < 0.05 a priori. To assess the frequency of implementation of the ICF checklist, measures of central tendency (mean, median, mode, and standard deviation) were calculated for each item on the checklist. 

## 3. Results

### 3.1. Participants

In total, 270 athletic trainers started the survey and 185 completed the survey: a completion rate of 68.5%. The participants shared their age, highest degree earned, number of years credentialed as an athletic trainer, race/ethnicity, current job setting, and number of patients or potential patients at their current job/clinical site. Most athletic trainers who participated (age = 35 ± 9 y; women = 122, 65.9%) had a professional (*n* = 82, 44.3%) or post-professional (*n* = 71, 38.4%) master’s degree, and 11 ± 8 years of experience as a credentialed athletic trainer. Table 1 describes the participants included in the analysis.

### 3.2. Familiarity and Knowledge

Over half of athletic trainers stated they were unfamiliar with the concept of disablement models (*n* = 96, 51.9%) and that they were unfamiliar with the ICF framework specifically (*n* = 99, 53.5%). Figure 1 describes the familiarity of the participants in more detail.

Most participants (*n* = 96, 51.9%) reported never learning about the ICF framework. Of the participants who had learned about the ICF framework, most were taught during their professional or post-professional athletic training education program (*n* = 48, 25.9%). Table 2 details the learning methods for the participants who had learned about the ICF framework.

The comparison of years of experience, relative to when the profession adopted the ICF disablement framework, revealed significant differences between groups (U = 3313, Z = −2.104, *p* = 0.035), whereby those with 0 to 7 years of experience (mean = 60.6 ± 30.3%) scored 18.3% higher than those with more than 7 years of experience (mean = 49.5 ± 35.0%).

On the knowledge assessment, the participants scored a 4.3 ± 2.7, which is equivalent to 53.7%. The components that over 60% of athletic trainers were able to correctly identify included Body Structure Impairments (*n* = 112, 60.5% correct), Activity Limitations (117, 63.2% correct), and Body Structures (133, 71.9% correct). The athletic trainers on the knowledge question related to *Participation Restrictions* (*n* = 28, 15.1% correct). Table 3 outlines the questions, correct answers, and number of participants who answered correctly for the knowledge assessment.

### 3.3. Sorting Application

During the sorting assessment, athletic trainers scored a 10.9 ± 3.9 out of 18, which is equivalent to 60.5%. The participants were best able to sort examples that correlated with body function and health conditions. However, participants inadequately sorted components into ICF categories for social cues in relationships into the activity component (*n* = 20, 10.8%) and living with family into the personal component (*n* = 26, 14.1%). Table 4 describes the sorting assessment and the number of correctly answered examples.

### 3.4. Implementation and Use

On the implementation matrix, the ICF tasks the athletic trainers most commonly elicited or recorded information about during patient care included neuromusculoskeletal movement (*n* = 65, 35.1%; body function), structure related to movement (*n* = 63, 34.1%, body structure), and mobility (*n* = 45, 24.3%; activity and participation). The most common never-elicited ICF tasks included voice and speech (*n* = 77, 41.6%; body function), sexual orientation (*n* = 71, 38.4%; personal), structure related to genitourinary and reproductive system (*n* = 51, 27.6%, body structure), and community, social, and civic life (*n* = 44, 23.8%; activity and participation). Table 5, Table 6, Table 7, Table 8 and Table 9 provide data about the participants patient load, and, when relevant to the patient’s condition, how often they elicit and/or document information on the functioning and disability of a patient specific to each ICF major component.

## 4. Discussion

Our study assessed athletic trainers’ familiarity, knowledge, application, and implementation of the ICF framework. It was important to explore these variables in athletic trainers, as the professional Strategic Alliance, including the NATA and the CAATE, adopted the ICF framework in 2015 as a foundation to guide patient evaluation [6]. The data collected suggest that ICF framework familiarity, knowledge, application, and implementation deficits exist for athletic trainers. While some athletic trainers reported use of the ICF framework in their practice, the overall implementation across all ICF major categories was low. The decision to not elicit information on these areas of health, regardless of reason, may reduce the ability to provide patient-centered, whole-person healthcare.

### 4.1. Familiarity with and Knowledge of Disablement Models

Disablement models are conceptual frameworks that can be used to guide patient evaluation, treatment, and overall care [15]. Their broad view of the patient, from injury diagnosis to impact on life roles, provides an opportunity to care for the patient in a more patient-centered manner. Patient-reported outcome measures are one tool available to address various disablement model components, yet familiarity with disablement models is low [16]. Specifically, clinicians with five or more years of experience were much less likely to be familiar with the ICF framework [17].

Our findings suggest that most athletic trainers reported being unfamiliar with or never learning about disablement models, much like other experienced healthcare clinicians. One contributing factor for this low awareness may be the reality that disablement models are relatively recent in athletic training, and adoption of the ICF has only been implemented since 2015. Additionally, the most recent educational standards, introduced in 2020, were the first to include the disablement model as a required element of professional education programs. Our data suggest that athletic trainers credentialed since 2015 (7 or fewer years of experience) scored significantly higher than those with more experience. This idea is supported by the work of Landon et al., who reported that clinicians with five or more years of experience were much less likely to be familiar with the ICF framework [16]. Research suggests that more direct experience with a topic results in greater familiarity with and understanding of said topic [18]. In our study, only 26% of participants had learned about the ICF framework through an athletic training program and 13.5% learned about it through a continuing education session. Low familiarity with a topic leads to decreased knowledge of that topic. In a similar study with occupational therapists and physiotherapists, only 29% of the respondents knew what the components of the ICF framework were [19]. Similarly, another study exploring healthcare professionals in developing countries identified that only 31% of respondents had knowledge of the ICF components [20]. Much like other healthcare professionals who have low familiarity on the topic of the ICF framework, athletic trainers also demonstrated low knowledge of the specific ICF framework components.

Participants who were enrolled in education programs after the framework was introduced and adopted were more likely to have been exposed to these models as part of their professional preparation school; however, this leaves many athletic trainers without professional development on the topic. While one may assume that experience will lead to exposure to emerging ideas that align with advancements in care, one study on physicians found the opposite was true, demonstrating the need for some sort of intervention for healthcare providers [21]. The field of athletic training is continually evolving, which means that mechanisms or processes are needed to ensure that all professionals have ways to gain information that aligns with new standards and advancements in practice. For example, the Board of Certification is evolving their programs and advancing continuing professional certification programs [22]. One option could be holding workshops for athletic trainers to attend, detailing the new standards and the scope of training expansion areas. In order to accomplish this, athletic trainers that were eligible for their Board of Certification exam prior to the new curricular standards and the scope of training expansions for athletic training, including the ICF frameworks, should have mandated training. The development of a competence assessment module focused on the ICF framework could serve as a meaningful learning opportunity to identify their need for self-actualized learning [22]. Furthermore, attendance at such workshops does not ensure comprehension or implementation. The continued professional development of athletic trainers surrounding new curricular content standards should be evaluated through knowledge assessments and quality improvement implementation projects [22]. It is crucial to increase familiarity because there is a positive correlation between familiarity with the ICF framework and the use of it [17].

### 4.2. Application of the ICF Framework

Athletic trainers in our study demonstrated difficulty applying specific ICF framework component examples to the correct major category. The concern regarding being unable to perform this task alludes to the fact that while athletic trainers may feel they know what goes into whole-person healthcare, they fall back on the ideas that impairments are the issue and they do not broaden their perspectives. The difficulty linking these could lead to poor goal setting development by the athletic trainer, which is focused on the anatomical issues (e.g., lack of arm strength) rather than activity limitation or participation restriction (e.g., carrying groceries up the stairs or golfing with friends). It could also lead to poor documentation organization. One mechanism that could help athletic trainers sort and apply the ICF framework to clinical practice is the use of the ICF browser tool [23]. This tool could be used by athletic trainers to quickly look up ICF framework components and gain an understanding of where they are classified. Additionally, ICF core sets have been developed and could be useful for athletic trainers to use in clinical practice. The ICF core sets were developed as a systematic way to describe health conditions and with a minimalistic design to keep use of the model efficient [21]. A simple example used by the World Health Organization, applying a health condition to various levels of functioning utilizing the ICF framework, was the health condition of a spinal injury, with the impairment being paralysis, the activity limitation being incapable of using public transportation, and the participation restriction of no participation in religious activities due to no public transportation use [24]. Without integrating the ICF framework, a clinician may have approached this patient by treating and managing their chief concern, which could be paralysis. The focus of the goals and care plan would be their paralysis in terms of range of motion and strength. Using a disablement model, the clinician is able to identify key areas that may be specific, activity-based goals that are of importance to the patient’s life. Tools such as the Disablement in the Physically Active Scale and the Patient-Specific Functional Scale have been developed to assist athletic trainers when exploring the problems their patients face after injury [25,26]. These tools also allow the patient to be integrated through a shared decision-making process to identify important activities they are unable to perform or struggle to accomplish due to their health condition. The use of these scales and goal setting mechanisms allows the athletic trainer to address personal factors, such as anxiety and fear, as well as important activities such as attending religious services. The provided case example demonstrates the value of applying ICF framework components to specific situations to provide the best whole-person healthcare to a specific individual’s situation, which sees a person as more than their condition or impairments.

### 4.3. Implementation of the ICF Framework in Clinical Practice

In our survey, using the implementation matrix, the most reported ICF tasks the athletic trainers elicited included neuromusculoskeletal movement (35.1%; body function), structure related to movement (34.1%, body structure), and mobility (24.3%; activity and participation). These findings are unsurprising, given the significant emphasis on musculoskeletal care and movement assessments that are part of athletic training education. The most common never-elicited ICF tasks included voice and speech, sexual orientation, structure related to genitourinary and the reproductive system, and community, social, and civic life. Although athletic trainers successfully incorporated the documentation relating to physical and biomechanical components, they struggled to implement the majority of the other ICF tasks, such as sexual orientation. It is important to increase the implementation of ICF components because it facilitates whole-patient care and allows interventions to be refocused to address the unique needs of each patient. Athletic trainers strongly believe that they are patient centered and have a willingness to be more patient centered, yet do not implement things such as the ICF framework into clinical practice [12]. This is why it is important to provide resources or educational opportunities for athletic trainers to learn more about certain components of the ICF framework. The survey indicated a weakness in asking about voice and speech, sexual orientation, or reproductive systems, and about community, social, and civic life. It is possible that athletic trainers work with a population in which these problems are not as common; however, if they do, athletic trainers could work interprofessionally with a speech language pathologist to better understand the importance of asking about voice and speech. Educational resources could be provided to athletic trainers on the patient-centered care of lesbian, gay, bisexual, transgender, and queer patients, because they have indicated a desire to learn how to provide the best care for that population [27]. Finally, attending events in the community to better understand the social and civic life of the patient population could be helpful, since community and social life can affect health outcomes [28,29].

### 4.4. Limitations and Future Research

While the data provide insight into the current knowledge and implementation of the ICF framework in athletic training, our study is not without limitations. First, the study included 185 participants, which equals a 2.64% response rate. This is relatively low. In addition, most of the sample was largely White/Caucasian, which means the results may not be generalized to other populations. However, it is indicative of the athletic training profession demographics. It is important to note that when conducting survey research, it is assumed that individuals are answering the questions truthfully and to the best of their ability, without additional resources. The study was limited to athletic trainers working with patients/in patient care. The results may have been reflected differently for athletic trainers working in research or healthcare administration. In addition, the purpose of the study was only to explore knowledge and implementation of the profession-adopted ICF framework and not to explore other disablement models (e.g., the Nagi model). Researchers can build active educational interventions with outcomes focused on improving the clinician’s familiarity and knowledge of the ICF framework. Further explorations of the most effective ways to support the implementation of the ICF framework into practice, including the study of simulation-based learning and the use of quality improvements projects, may help to develop the educational intervention for a clinician to understand what parts of the ICF framework apply to different aspects of the examination.

## 5. Conclusions

While some athletic trainers are incorporating the ICF model into practice, many retain deficits related to familiarity, knowledge, application, and implementation of the ICF framework into clinical practice. Athletic trainers reported low implementation across all ICF categories. The decision not to elicit information on these areas of health, through use of the ICF framework, is a missed opportunity to support the delivery of patient-centered, whole-person healthcare.

## Figures and Tables

**Figure 1 ijerph-20-05401-f001:**
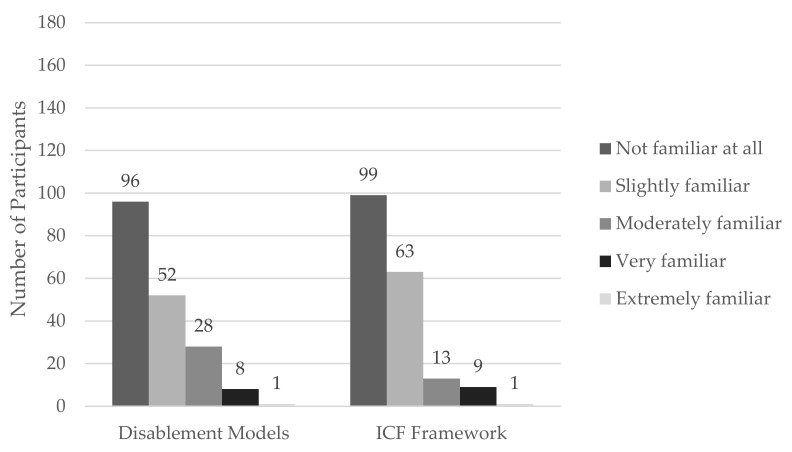
Participant familiarity with Disablement Models and the ICF framework.

**Table 1 ijerph-20-05401-t001:** Participant Demographics.

Characteristic	*n*, %
Gender	
Woman	122, 65.9%
Man	60, 32.4%
Transgender Man	1, 0.5%
Prefer to Self-Describe	2, 1.1%
**Ethnicity**	
White/Caucasian	159, 85.9%
American Indian/Alaskan Native/Indigenous	1, 0.5%
Asian/Asian American	1, 0.5%
Black/African American	5, 2.7%
Hispanic/Latinx	5, 2.7%
Native Hawaiian/Pacific Islander	1, 0.5%
2 or more races	9, 4.9%
Another option not listed	1, 0.5%
Prefer not to say	3, 1.6%
**Job Setting**	
Amateur, Recreation or Youth Sports	2, 1.1%
Clinicand Physician Practice	15, 8.1%
College/University	77, 41.6%
Health/Fitness/Sports/Performance Enhancement Hospital	2, 1.1%
Independent Contractor	1, 0.5%
Military, Law Enforcement and Government	1, 0.5%
Occupational Health and Industrial	9, 4.9%
Performing Arts	12, 6.5%
Professional Sports	3, 1.6%
Secondary School	5, 2.7%
	58, 31.4%

**Table 2 ijerph-20-05401-t002:** Participant educational experiences related to the ICF framework.

Learning Method	*n*, %
I have not learned about it	96/185, 51.9%
I have learned about it (Select all that apply)	89/185, 48.1%
During my Professional Athletic Training Program	24
During my Post-Professional Athletic Training Program	24
Informally by a colleague	18
Informally through self-education	18
Continuing education session by an athletic trainer	17
Reading journal articles	14
Continuing education session by a different healthcare provider	8
Required/Mandated in-service by employer	6
Other	5

**Table 3 ijerph-20-05401-t003:** Participant knowledge scores related to the different components of the ICF framework.

Question stem: What Component of the ICF Framework Does the Following Example Describe?	Correct Answer	Participants Answering Correctly (*n*, %)
Physiological functions of body systems	Body Function	106, 57.3%
The execution of a task or action by an individual	Activity	97, 52.4%
Problems an individual may experience during involvement in life situations	Participation Restrictions	28, 15.1%
Problems in body structure such as significant deviation or loss	Body Structure Impairments	112, 60.5%
Anatomical parts of the body such as organs, limbs, and their components	Body Structures	133, 71.9%
The physical, social, and attitudinal environment in which people live and conduct their lives	Environmental Factors	111, 60.0%
Difficulties an individual may have in executing tasks like lifting weights or combing hair	Activity Limitations	117, 63.2%
Involvement in a life situation	Participation	91, 49.2%

**Table 4 ijerph-20-05401-t004:** Sorting Application Matrix of Category Statements.

ICF Framework Component	Correct ICF Framework Major Category	Correctly Sorted (*n*, %)
Regulation of emotion	Body function	44, 23.8%
Managing diet and fitness	Activity	85, 45.9%
Individual attitudes of friends	Environmental	86, 46.5%
Muscles of shoulder region	Body structure	141, 76.2%
Generalized anxiety disorder	Health condition	122, 65.9%
Abuse of addictive drugs	Personal	42, 22.7%
Heart rate	Body function	131, 70.8%
Bursitis related to use, overuse, and pressure	Health condition	119, 64.3%
Cornea	Body structure	141, 76.2%
Population density	Environmental	141, 76.2%
Living with family	Personal	26, 14.1%
Managing daily routine	Activity	115, 62.2%
Acute Sinusitis	Health condition	141, 76.2%
Social cues in relationships	Activity	20, 10.8%
Immediate family	Environmental	74, 40.0%
Female biological sex	Personal	52, 28.1%
Power of muscles in lower half of body	Body function	121, 65.4%
Kidney	Body structure	140, 75.7%

**Table 5 ijerph-20-05401-t005:** Participant Report of Screening in Clinical Practice for Impairments of Body Functions (*n*, %).

Impairments	Never Elicited	Some Patients	About Half of My Patients	Most of My Patients	Always Elicited	Did Not Answer
Mental (ex: sleep, orientation, emotion)	19, 10.3%	70, 37.8%	26, 14.1%	30, 16.2%	11, 5.9%	29, 15.7%
Sensory (ex: seeing, hearing, pain)	25, 13.5%	39, 21.1%	12, 6.5%	44, 23.8%	36, 19.5%	29, 15.7%
Voice and Speech	77, 41.6%	55, 29.7%	2, 1.1%	16, 8.6%	5, 2.7%	30, 16.2%
Cardiovascular, Hematological, Immunological, and Respiratory Systems (ex: heart, blood pressure, breathing, allergies)	20, 10.8%	70, 37.8%	22, 11.9%	29, 15.7%	14, 7.6%	30, 16.2%
Digestive, Metabolic, and Endocrine Systems (ex: weight, hormones, defecation)	41, 22.2%	75, 40.5%	19, 10.3%	14, 7.6%	6, 3.2%	30, 16.2%
Genitourinary and Reproductive (ex: urinary, sexual)	67, 36.2%	67, 36.2%	10, 5.4%	9, 4.9%	1, 0.5%	31, 16.8%
Neuromusculoskeletal and Movement (ex: mobility, power, tone)	9, 4.9%	20, 10.8%	7, 3.8%	54, 29.2%	65, 35.1%	30, 16.2%
Skin and Related Structures	19, 10.3%	55, 29.7%	32, 17.3%	37, 20%	12, 6.5%	30, 16.2%

**Table 6 ijerph-20-05401-t006:** Participant Report of Screening in Clinical Practice for Impairments of Body Structures (*n*, %).

Impairments	Never Elicited	Some Patients	About Half of My Patients	Most of My Patients	Always Elicited	Did Not Answer
Nervous System structures (ex: brain, spinal cord, nerves)	11, 5.9%	56, 30.3%	24, 13%	29, 15.7%	24, 13%	41, 22.2%
Eye, ear, and related structures	14, 7.6%	78, 42.2%	20, 10.8%	19, 10.3%	14, 7.6%	40, 21.6%
Structures involved in voice and speech	49, 26.5%	72, 38.9%	3, 1.6%	11, 5.9%	10, 5.4%	40, 21.6%
Structure of the cardiovascular, immunological, and respiratory systems	20, 10.8%	75, 40.5%	19, 10.3%	17, 9.2%	14, 7.6%	40, 21.6%
Structures related to the digestive, metabolism, and endocrine systems	37, 20%	73, 39.5%	14, 7.6%	13, 7%	8, 4.3%	40, 21.6%
Structures related to genitourinary and reproductive system	51, 27.6%	73, 39.5%	6, 3.2%	8, 4.3%	7, 3.8%	40, 21.6%
Structure related to movement (ex: shoulder, pelvis, lower extremity)	8, 4.3%	18, 19.7%	8, 4.3%	48, 25.9%	63, 34.1%	40, 21.6%
Skin and related structures	13, 7%	62, 33.5%	20, 10.8%	32, 17.3%	18, 9.7%	40, 21.6%

**Table 7 ijerph-20-05401-t007:** Participant Report of Screening in Clinical Practice for Activity Limitations and Participation (*n*, %).

Limitation	Never Elicited	Some Patients	About Half of My Patients	Most of My Patients	Always Elicited	Did Not Answer
Learning and Applying Knowledge (ex: listening, watching, solving problems)	27, 14.6%	56, 30.3%	17, 9.2%	21, 11.4%	18, 9.7%	46, 24.9%
General Tasks and Demands (ex: single tasks, multiple tasks)	26, 14.1%	46, 24.9%	21, 11.4%	28, 15.1%	19, 10.3%	45, 24.3%
Communication (ex: spoken message, non-verbal message, conversation)	28, 15.1%	50, 27%	19, 10.3%	29, 15.7%	14, 7.6%	45, 24.3%
Mobility (ex: lifting, carrying, grasping, driving)	9, 4.9%	21, 11.4%	18, 9.7%	47, 25.4%	45, 24.3%	45, 24.3%
Self Care (ex: washing, toileting, dressing, eating, drinking)	28, 15.1%	57, 30.8%	15, 8.1%	24, 13%	14, 7.6%	47, 25.4%
Domestic Life (ex: shopping, cooking, cleaning, assisting others)	42, 22.7%	46, 24.9%	18,9.7%	26, 14.1%	8, 4.3%	45, 24.3%
Interpersonal interactions and relationships (ex: basic, complex, strangers, social, family, intimate)	40, 21.6%	63, 34.1%	19, 10.3%	11, 5.9%	6, 3.2%	46, 24.9%
Major life areas (ex: school, higher education, employment, economic self-sufficiency)	28, 15.1%	53, 28.6%	17, 9.2%	30, 16.2%	11, 5.9%	46, 24.9%
Community, social, and civic life (ex: recreation, leisure, religion, spirituality, political life)	44, 23.8%	47, 25.4%	18, 9.7%	22, 11.9%	8, 4.3%	46, 24.9%

**Table 8 ijerph-20-05401-t008:** Participant Report of Screening in Clinical Practice for Environmental Factors (*n*, %).

Factor	Never Elicited	Some Patients	About Half of My Patients	Most of My Patients	Always Elicited	Did Not Answer
Products and technology (ex: food, medicine, transportation)	22, 11.9%	55, 29.7%	21, 11.4%	34, 18.4%	4, 2.2%	49, 26.5%
Natural environment and human made changes to environment (ex: climate, light, sound)	31, 16.8%	66, 35.7%	19, 10.3%	18, 19.7%	2, 1.1%	49, 26.5%
Support and relationships (ex: immediate family, friends, neighbors, health professionals)	19, 10.3%	63, 34.1%	23, 12.4%	27, 14.6%	4, 2.2%	49, 26.5%
Attitudes (ex: individual attitudes of friends, societal attitudes, social norms, ideologies)	23, 12.4%	61, 33.0%	27, 14.6%	22, 11.9%	3, 1.6%	49, 26.5%
Services, systems, and policies (ex: housing, transportation, education, health services, legal, social security)	30, 16.2%	69, 37.3%	18, 9.7%	16, 8.6%	3, 1.6%	49, 26.5%

**Table 9 ijerph-20-05401-t009:** Participant Report of Screening in Clinical Practice for Personal Factors (*n*, %).

Factor	Never Elicited	Some Patients	About Half of My Patients	Most of My Patients	Always Elicited	Did Not Answer
Lifestyle	21, 11.4%	56, 30.3%	20, 10.8%	28, 15.1%	9, 4.9%	51, 27.6%
Habits	21, 11.4%	53, 28.6%	24, 13.0%	28, 15.1%	8, 4.3%	51, 27.6%
Social Background	32, 17.3%	70, 37.8%	12, 6.5%	15, 8.1%	5, 2.7%	51, 27.6%
Education	43, 23.2%	54, 29.2%	8, 4.3%	23, 12.4%	6, 3.2%	51, 27.6%
Life Events	16, 8.6%	65, 35.1%	20, 10.8%	26, 14.1%	7, 3.8%	51, 27.6%
Race/ethnicity	59, 31.9%	48, 25.9%	5, 2.7%	16, 8.6%	6, 3.2%	51, 27.6%
Sexual orientation	71, 38.4%	51, 27.6%	4, 2.2%	7, 3.8%	1, 0.5%	51, 27.6%
Individual psychological assets	31, 16.8%	68, 36.8%	15, 8.1%	17, 9.2%	3, 1.6%	51, 27.6%
Age	27, 14.6%	45, 24.3%	12, 6.5%	18, 9.7%	32, 17.3%	51, 27.6%
Gender	37, 20.0%	42, 22.7%	7, 3.8%	21, 11.4%	27, 14.6%	51, 27.6%
Upbringing	55, 29.7%	56, 30.3%	12, 6.5%	11, 5.9%	0, 0.0%	51, 27.6%
Food Preferences	40, 21.6%	59, 31.9%	13, 7.0%	20, 10.8%	2, 1.1%	51, 27.6%
Fitness	11, 5.9%	41, 22.2%	18, 9.7%	39, 21.1%	25, 13.5%	51, 27.6%
Coping Style	36, 19.5%	48, 25.9%	28, 15.1%	18, 9.7%	4, 2.2%	51, 27.6%

## Data Availability

The data presented in this study are available on request from the corresponding author. The data are not publicly available due to IRB protections.

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
