# Peer review of "The Knowledge and Use of the International Classification of Functioning, Disability and Health (ICF) Framework in Athletic Training"

_ijerph, 2023, doi:10.3390/ijerph20075401_

Round 1

Reviewer 1 Report

Dear authors,

Thank you for submitting your manuscript for review. Please find below my commentaries.

Line 42-45: the same phrase/idea is repeated: ICF is a disablement model that helps clinicians to focus on the patients unique needs.

Table 1: the total sample was is 185. However, the sum of individuals listed by ethnicity is 195. If only one answer was possible (considering that “2 of more races” was an option”), an error must exist here. Also, the sum of percentages is 105.3%, which can be only due to rounding error.

Line 125: more information on how the content validation index was obtained is advised. The authors should be clear on how this was performed (which content analysis methods were used), or if only the evaluation of the experts were taken in consideration.

Line 157: according to Figure 1 (and the text on lines 155-156), the number of participants who may have answered as “never learning about ICF” should be 99, and not 96, as stated in line 15.

Line 160: there seems to be an issue with the percentage of people who were taught about ICF during their athletic training education program. The percentage is not right if n=48 is gathered from the total sample (48/185 = 25.9%), or from those that knew about ICF [(185-99)/185 = 46,5%]. Also, for n=48 be 20.9%, the sample size would have to be 230. However, after reading Table 2 and the discussion, I became aware the authors were including the participants that learned ICFs during the professional (24) or post-professional athletic training (24). This kind of confusion can also occur with the reader. The authors should consider how the text (lines 155-172) relates with the supporting Figures (1) and Tables (2). In line 155-158 the data is presented in Figure 1, but the information of lines 158-160 requires consultation of Table 2, which has not yet been introduced in the text (is only introduced in the next paragraph).

Table 2: again, this is confusing. According to Figure 1, 96 participants are not familiar at all with disablement models. Also, 99 are not familiar at all with the 99 ICF framework. However, Table 2 stated that 96 participants have not learned about ICF. Since this lack of knowledge is related to the ICF framework, shouldn’t table 2 report n = 99?

Table 2: Similarly, the sum of the remaining options is 134, which with the 96 other participants that answered with “I have not learned about it”, would total 230. Is the second time I get this total number (see comment about line 160). However, the first line of the table states that 96 corresponds to 51.9%, which is line with a sample size of 185 participants. Were the participants allowed to select more than one answer? I believe this was not disclosed on the methods.

Line 327: since the authors state that the use of the ICF browser tool would be helpful for practitioners, I would advise to include a reference to here this tool may be accessed.

Considering that this study is based on a questionnaire, it would be useful for readers and anyone thinking in perform a similar or replication study, to have access to the questionnaire presented. I suggest including a copy of the questionnaire as supplemental material.

Author Response

Reviewer 1

Line 42-45: the same phrase/idea is repeated: ICF is a disablement model that helps clinicians to focus on the patients unique needs.

Response: This has been removed.

Table 1: the total sample was is 185. However, the sum of individuals listed by ethnicity is 195. If only one answer was possible (considering that “2 of more races” was an option”), an error must exist here. Also, the sum of percentages is 105.3%, which can be only due to rounding error.

Response: Thank you for the comment. We have revised for participants that selected 2 or more races into the 2 or more-race category resulting in 185 responses.

Line 125: more information on how the content validation index was obtained is advised. The authors should be clear on how this was performed (which content analysis methods were used), or if only the evaluation of the experts were taken in consideration.

Response: We have expanded on the CVI process. There is also a citation provided on the process.

Line 157: according to Figure 1 (and the text on lines 155-156), the number of participants who may have answered as “never learning about ICF” should be 99, and not 96, as stated in line 15.

Response: This data is correct. 96 ATs reported never learning about the disablement models. However, the data you are suggesting is wrong (n=99) is about how familiar they are with the ICF disablement model. There were 99 people stated they were not familiar with the ICF at all, which could suggest 3 people had learned about disablement models but were still unfamiliar with the ICF model. This is the reason for the noted discrepancy.

Line 160: there seems to be an issue with the percentage of people who were taught about ICF during their athletic training education program. The percentage is not right if n=48 is gathered from the total sample (48/185 = 25.9%), or from those that knew about ICF [(185-99)/185 = 46,5%]. Also, for n=48 be 20.9%, the sample size would have to be 230. However, after reading Table 2 and the discussion, I became aware the authors were including the participants that learned ICFs during the professional (24) or post-professional athletic training (24). This kind of confusion can also occur with the reader.

We agree with the change to 25.9% for the data in the text. This has been revised. We have revised Table 2 to note that participants could select more than 1 learning method.

The authors should consider how the text (lines 155-172) relates with the supporting Figures (1) and Tables (2). In line 155-158 the data is presented in Figure 1, but the information of lines 158-160 requires consultation of Table 2, which has not yet been introduced in the text (is only introduced in the next paragraph).

Response: We have revised this section in terms of noting Table 2 and Figure 1.

Table 2: again, this is confusing. According to Figure 1, 96 participants are not familiar at all with disablement models. Also, 99 are not familiar at all with the 99 ICF framework. However, Table 2 stated that 96 participants have not learned about ICF. Since this lack of knowledge is related to the ICF framework, shouldn’t table 2 report n = 99?

Response: While we respect your interpretation and agree, 96 participants stated they had not learned about this at all. The additional 3 participants indicating unfamiliarity with the ICF may have learned about it but still not familiar with it at all. This is typical of educational research with learning about something and familiarity with a topic are different constructs.

Table 2: Similarly, the sum of the remaining options is 134, which with the 96 other participants that answered with “I have not learned about it”, would total 230. Is the second time I get this total number (see comment about line 160). However, the first line of the table states that 96 corresponds to 51.9%, which is line with a sample size of 185 participants. Were the participants allowed to select more than one answer? I believe this was not disclosed on the methods.

Response: Thank you for the many comments on Table 2. We have revised it to be clearer. We have set this up to be learned/did not learn which is the 89 and 96. Then from the 89 who did learn, we have 134 responses of their select all that apply methods.

Line 327: since the authors state that the use of the ICF browser tool would be helpful for practitioners, I would advise to include a reference to here this tool may be accessed.

Response: We have provided a citation for the browser tool website.

Considering that this study is based on a questionnaire, it would be useful for readers and anyone thinking in perform a similar or replication study, to have access to the questionnaire presented. I suggest including a copy of the questionnaire as supplemental material.

Response: Survey has been added.

Reviewer 2 Report

The topic addressed in this manuscript might interest the International Journal of Environmental Research and Public Health readers. The study focused on current practicing athletic trainers in their knowledge and skills in applying the ICF model in their current practice. The study used a survey design to examine athletic trainers’ knowledge of the ICF model. The results found that many athletic trainers lacked knowledge about the ICF and were not able to apply the ICF in their practice of patient care.

Minor concerns

Introduction

1) The introduction could provide more information regarding the ICF model with the introduction of the different components of ICF. It will also benefit the readers if practical examples of how ICF is used in athletic training.

Method

1) It might be a good idea to provide the survey used in the study as appendix.

2) Thank you for providing the content validity evidence of the survey.

Results

1) Thank you for providing an easy-to-understand results section.

2) Results of the Mann-Whitney U were not presented.

Author Response

Reviewer 2

The introduction could provide more information regarding the ICF model with the introduction of the different components of ICF. It will also benefit the readers if practical examples of how ICF is used in athletic training.

Response: We have provided an example of the ICF use in AT.

It might be a good idea to provide the survey used in the study as appendix.

Response: Survey has been provided.

Results of the Mann-Whitney U were not presented.

Response: It appears this is presented in the results section 3.2.

Reviewer 3 Report

In summary, this is a cross-sectional survey design that investigated athletic trainers' familiarity, knowledge, application, and implementation of the ICF framework. The authors found that there were deficits in these among athletic trainers. While the study is limited by cross-sectional design, the authors seemed to do their best to increased the quality from the survey development phase. Results are well presented and intro/discussion are well written. I only have few comments seen below: 

Line 91-98 and Table 1: I would move this section to Results section. 

Line 130-137: Was there consent form/page at the start of survey where participants can click either agree or disagree to participate? any incentives given for participation?

Analyis/Results: Line 161-163. Any difference of scores among gender, job settings, degrees, or other factors? Also, was there difference in terms of implementation between the years of experience?

Limitations: the participant population was largely white/caucasian. The results may not be generalized to other populations. Also, very low response rate 185/7000. Please mention this in limitation and how it may potentially lead to bias. 

Please attach a sample of survey as a supplement. 

Author Response

Reviewer 3

Line 91-98 and Table 1: I would move this section to Results section. 

Response: Revised as suggested.

Line 130-137: Was there consent form/page at the start of survey where participants can click either agree or disagree to participate? any incentives given for participation?

Response: We have included more details on the consent process.

Analysis/Results: Line 161-163. Any difference of scores among gender, job settings, degrees, or other factors? Also, was there difference in terms of implementation between the years of experience?

Response: This was not the part of the aims of the present study. The gender of an individual would not impact the knowledge nor the continuing education efforts directed by the profession. In addition, the degree change in athletic training to the masters level skews the data for comparison which is why we evaluated years of experience since the introduction of the ICF to the profession.

Limitations: the participant population was largely white/Caucasian. The results may not be generalized to other populations. Also, very low response rate 185/7000. Please mention this in limitation and how it may potentially lead to bias. 

Response: Revised as requested.

Please attach a sample of survey as a supplement. 

Response: Survey has been provided.